# Threshold Electricity Consumption Enables Multiple Sustainable Development Goals

**Robert J. Brecha** [1,2,3]

1    Physics Department and Renewable and Clean Energy Program, University of Dayton, Dayton, OH 45469, USA; robert.brecha@climateanalytics.org
2    Hanley Sustainability Institute, University of Dayton, Dayton, OH 45469, USA
3    Climate Analytics, Ritterstrasse 3, 10969 Berlin, Germany

**Abstract:** Access to sufficient amounts of energy is a prerequisite for the development of human well-being. The Sustainable Development Goals (SDGs) recognize the interconnectedness of climate change, energy access and development. However, not all SDG targets are quantified, leaving room for ambiguity in fulfilling, for example, the goal of ensuring access to affordable, reliable, sustainable and modern energy for all (SDG7). We show how specific sustainable development targets for health indicators are strongly correlated with electricity consumption levels in the poorest of countries. Clear thresholds in per capita electricity consumption of a few hundred kWh per year are identified by analyzing SDG indicator data as a function of per capita country electricity consumption. Those thresholds are strongly correlated with meeting of SDG 3 targets-below the identified thresholds, countries do not meet the SDG targets, while above the threshold there is a clear relationship between increasing consumption of electricity and improvement of SDG indicators. Electricity consumption of 400 kWh per capita is significantly higher than projections made by international agencies for future energy access, but only 5%–10% that of OECD countries. At the very least, the presence of thresholds and historical data patterns requires an understanding of how SDG targets would be met in the absence of this threshold level of electricity access.

**Keywords:** sustainable development; energy access; climate change; renewable energy; electricity

## 1. Introduction

Historically, human development has been strongly correlated with energy consumption [1–3]. That there is also causation in the relationship between energy access and development, rather than simply correlation, is implicitly acknowledged by the many international efforts to increase access to modern energy systems in developing countries [4–6], as well as being explicitly recognized in the United Nations Sustainable Development Goals (SDGs): SDG7 is to "ensure access to affordable, reliable, sustainable and modern energy for all" [7].

Very broadly speaking, SDG7 considers through Target 7.1 the need to "By 2030, ensure universal access to affordable, reliable and modern energy services." What is meant by modern energy is shown in the indicators corresponding to Target 7.1, *i.e.,* measurement of the "Proportion of population with access to electricity" and "Proportion of population with primary reliance on clean fuels and technology." In particular, for the first of these indicators, there are two dimensions to the measurement that are important: First, the issue of access to the modern energy source itself, and second, what level of actual consumption is significant and for which purposes. Quantifying more precisely what "access" to energy means is the subject of an extensive literature [8–15]. One approach is to consider energy consumption in three "locales" to define energy access: "households," "productive engagements" and "community facilities" [16]. Within these locales, different tiers of energy access, for different types of

energy (electricity for various activities, cooking fuels, space heating) are defined, with several different tiers and indices formulated for each. While specifics differ across locales, one common denominator is the importance of access to and consumption of electricity, as the most flexible energy source; this is also the focus of the present work.

The International Energy Agency (IEA), in highlighting the critical need for modern energy to enable development, defines for individual households (as only one component of electricity consumption) initial electricity "access" as 250 kWh per rural household per year, or 500 kWh for an urban household (assumed to be five persons) [5,17] with increasing consumption over time to reach the country or regional average level of consumption. This level of consumption corresponds to approximately Tier 3 (out of five) access in Reference [16]. The present work connects to the literature that goes beyond households since much of the critical consumption of electricity necessary for meeting SDGs will be outside the home, in schools and clinics and commercial enterprises, for example. Because data are often not available at sufficient sectoral detail for many countries, and especially so in the case of the least-developed countries (LDCs) as classified by the World Bank, the countries with lowest energy consumption and lowest rates of access, we rely here on aggregated data. Specifically, due to the focus on electricity, we use annual electricity consumption per capita in a country as the chief independent variable for analysis.

Just as the bare fact of access (versus lack of access) to energy is not sufficient for guaranteeing development outcomes, neither is a given level of energy consumption. Details of how energy access and consumption in a country are distributed among different populations is crucially important as well. In addition, how energy consumption is allocated in different sectors plays an important role—whether energy is used for directly productive social purposes such as community facilities, as opposed to, for example supporting extractive export industries. Finally, the physical sources of modern energy are important in connection to other sustainability criteria, most notably that of mitigating the worst impacts of climate change, which therefore implies a preponderance of carbon-free sources of energy. Implicit in the present work is that "electricity" means "sustainably-sourced electricity," and that the other factors just described will also be part of the implementation of modern energy access. With these caveats in mind, the focus here is on aggregate indicators of electricity consumption on a national level and the identification of minimal acceptable levels of consumption based on historical data and correlations.

There is also an increasing literature on assessing SDGs and their various interactions [18–21]. In particular, McCollum et al. [21] explicitly link energy targets to other SDGs in a qualitative literature review of synergies (and in fewer cases, potential conflicts between goals). Although not available for all SDGs or all targets within the SDGs, whenever possible it would be useful to have quantitative targets to use as measurable indicators for progress toward reaching the SDGs, while recognizing the fact that a wide variety of development pathways have been followed over time. The approach used here is to identify indicators for SDG targets of the form, "to date, only a very small percentage of countries with less than $x$ amount of energy meet SDG target $y$." Even if it is not necessarily possible to construct a seamless chain of causality between $x$ and $y$, interest in enabling countries achieve SDG targets requires at least an explanation of how the historical norm would be circumvented. Therefore, we examine historical indicator data relevant to a number of SDGs and correlate achieving these goals and targets with the amount of average per capita yearly electricity consumption in a given country. We describe our data sources and methods in Section 2. The main body of the paper consists of a presentation of correlations between electricity access and specific SDG targets, primarily related to health (Section 3.1). In addition, thresholds are identified in electricity consumption that relate to SDG targets. In Section 3.2 an estimate of electricity access needs that are consistent with achieving the SDGs in Sub-Saharan Africa is presented, and conclusions and discussion are given in Section 4.

## 2. Data and Methods

Data for SDG indicators are taken from the World Bank database of World Development Indicators (WDI) [22]. Population data, both current and projected increases, are taken from the United Nations [23]. Electricity data were sourced from the US Energy Information Administration (EIA) [24]. The United Nations Environment Programme *Atlas of Africa Energy Resources* was used [25] as a cross-check for countries where data availability is scarce. In total, data from 179 countries were available for the main analysis presented here.

The approach taken here is to first identify correlations between electricity consumption and health indicators, to look for significant thresholds in those correlations and then to examine distributions of countries having either less than, or more than, these threshold values of per capita electricity consumption, to determine how these countries fare in meeting SDG numerical targets. Here we concentrate on health indicators, which show the clearest threshold behavior. For each group a histogram of indicator values was created. Specific targets stated in the SDGs were used for binning histograms, such that countries already meeting those targets and those yet to do so are clearly separated.

We choose a logarithmic scale for both axes to allow for an interpretation of the slope as an "elasticity;" that is, we can treat the slope of the graph as giving information about the fractional change in the independent variable that gives rise to a resulting fractional change in the dependent variable, and a least-squares fitting routine is used to both identify any existing breakpoint in the data, and to then determine the slopes both before and after the breakpoint [26]. Using logarithmical scales is common in the economics literature, but here we treat energy consumption as the independent variable. In general terms, if the relationship between two variables can be described by a power law of the form $Y = aX^b$, then taking logarithms of both sides leads to the equation $\ln(Y) = \ln(a) + b \ln(X)$. Plotting this equation allows the coefficient (power) to be determined. Furthermore, we can show by taking derivatives of both sides of the second equation that the interpretation of the power coefficient is $b = (\Delta Y/Y)/(\Delta X/X)$, i.e., that a fractional change (x%) in the variable X will result in a fractional change (y%) in the variable Y, scaled by a factor given by $b$.

## 3. Electricity Access for Sustainable Development

The key aim of this paper is to investigate the correlation between *electricity* consumption, as one component of energy availability, and the Sustainable Development Goals. Special attention will be given to the interaction between SDG7, Energy Access, and SDG3, Health. In the Supplementary Information we summarize some well-known relationships between the Human Development Index (HDI) and its components, and per capita annual energy and electricity consumption. The general feature is that of significantly increasing HDI with increasing electricity consumption at lower HDI levels, with a saturation in increase at higher levels. While one of the components of HDI is related to health (life expectancy at birth), in the next sub-sections we look more specifically at SDG 3 targets that contribute to human development and how these targets correlate with per capita electricity consumption. Also in the SI we look at similar connections between electricity consumption and other SDGs.

The focus here is on the interaction between SDG7 and SDG3 due to clear pathways linking access to energy and improved health outcomes. [27,28]. In particular, the first two targets for SDG3, to "Ensure healthy lives and promote well-being for all at all ages," which will be quantitatively described below, involve maternal, neo-natal, and infant mortality rates. In terms of energy access, and more accurately here, different levels of energy consumption, we choose to examine specifically electricity use, being the most flexible and useful energy source in the context of community facilities. For example, electric lighting in clinics allows better care access at all times of the day or night. Reliable and sufficient electricity supply allow the regular use of sterilization and diagnostic equipment, as well as emergency equipment needed for newborns and their mothers. Likewise, having access to computers and to communications equipment (connecting homes and clinics or hospitals) can improve birth and neo-natal outcomes. Lack of electricity for water supplies has also been indicated as a reason

for compromised birth and neo-natal services. [29] For infants, refrigeration can help in preserving vaccines, leading to lowered susceptibility to preventable diseases. This very brief indicative survey motivates the initial look at correlations between electricity consumption and SDG3 targets.

### 3.1. Sustainable Development Goals and Electricity Consumption

SDG7, Target 1 refers to energy access, without further more detailed quantification. Likewise, SDG7, Target 2, "By 2030, increase substantially the share of renewable energy in the global energy mix" does not provide further specificity of what "substantial" means. Only SDG7, Target 3 is more specific, "By 2030, double the global rate of improvement in energy efficiency" in terms of primary energy and GDP. Other SDGs have defined quantitative targets that can be measured more precisely. Targets associated with SDG3, are "By 2030, reduce the global maternal mortality ratio to less than 70 per 100,000 live births," and "By 2030, end preventable deaths of newborns and children under 5 years of age, with all countries aiming to reduce neonatal mortality to at least as low as 12 per 1000 live births and under-5 mortality to at least as low as 25 per 1000 live births" [7].

In Figure 1a we plot maternal mortality rate vs. electricity consumption per capita for 140 countries. As described in the Data and Methods section, we choose logarithmic scales; in the SI we present the data on linear axes for reference. A clear threshold electricity consumption is found at 280 kWh/capita and the slope is for higher consumption above the threshold is −1.04.

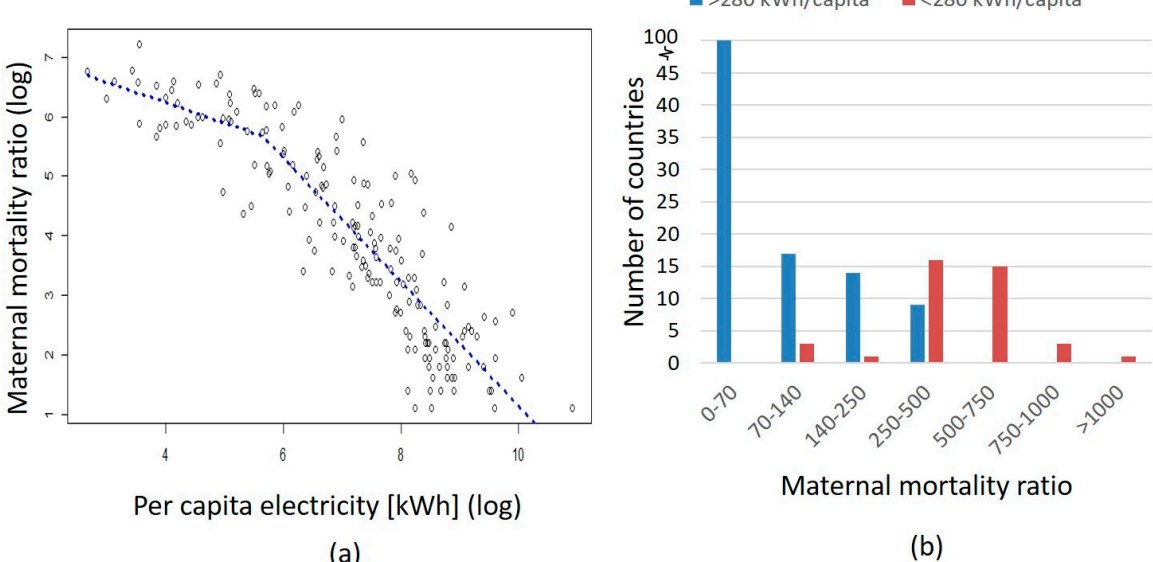

(a)

(b)

**Figure 1.** (**a**) Threshold fitting of maternal mortality ratio (log) vs. per capita national electricity consumption (log); (**b**) Histograms showing that for none of the countries with less than the indicative threshold electricity consumption, the SDG3 target of reducing maternal mortality rates to below 70 per 100,000 live births is met. The bin widths are chosen to emphasize the SDG numerical targets.

The interpretation can be made that average annual per capita societal electricity consumption of 280 kWh represents a threshold for improvement in neonatal mortality rates. Above this level, given the slope of −1.04 and the log-log form, there is an "elasticity" of improvement in maternal mortality as a function of electricity consumption (again, societal per capita amount) such that a 10% increase in electricity corresponds to a 10.4% decrease in maternal mortality. Regression parameters are shown in the Supplementary Information. A separate point of analysis is to ask if the two straight-line fits could, in fact, be explained by one linear fit. In the case shown here, the two slopes do not show overlap of the 95% confidence intervals.

With a potential threshold identified, the same data can be examined in terms of histograms of countries grouped on the basis of electricity consumption, and then binned based on SDG indicators,

as shown in Figure 1b. To interpret Figure 1b, we note the crucial point that there is no country with average electricity consumption below the threshold value of 280 kWh/capita that has a maternal mortality rate of less than 70 per 100,000 live births (and very few, with even two or three times that level). Conversely, if a country does have consumption to at least 280 kWh per capita of electricity, the likelihood of high maternal mortality is significantly lower (40 out of 140 countries).

Turning to the SDG3 target of reducing neonatal mortality rates, as shown in Figure 2a, a threshold can again be identified using the same method, here at 400 kWh/capita, representing a threshold for improvement in neonatal mortality rates with an elasticity of improvement in neo-natal mortality as a function of average per capita electricity consumption such that a 10% increase in electricity corresponds to a 7.2% decrease in neonatal mortality. Regression parameters are again shown in the Supplementary Information. The two slopes do not show overlap of the 95% confidence intervals.

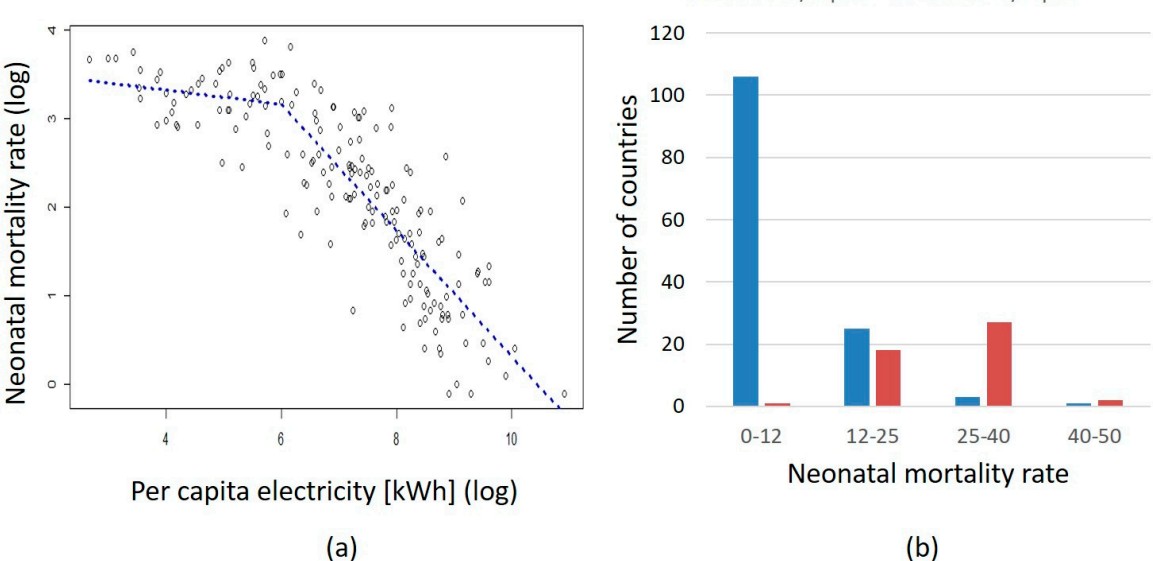

**Figure 2.** (**a**) Threshold fitting of neonatal mortality ratio (log) vs. per capita national electricity consumption (log); (**b**) Histograms showing that for only one of the countries with less than the indicative threshold electricity consumption of 400 kWh/capita, the SDG3 target of reducing maternal mortality rates to below 12 per 1000 live births is met.

In terms of a histogram, the interpretation of Figure 2b is that only one country with less than the threshold value of 400 kWh/capita (red bars) meets the SDG target of <12/1000 neonatal mortality, but that relatively few countries (29 out of 135) with greater than 400 kWh do *not* achieve the SDG target.

The third target under SDG3 is to reduce under-five mortality. The same breakpoint fitting procedure can be applied to other health indicators. In this case the threshold is found to be at a 350 kWh/capita, and the slope is –0.74, again to be interpreted as an elasticity, with the results shown in Figure 3a. The two slopes do not show overlap of the 95% confidence intervals.

The specific SDG3 target is to reduce under-5 mortality rates to below 25/1000. In Figure 3b a histogram binned to reflect the threshold found from the regression fit shows that no country with less than 350 kWh/capita electricity consumption meets the SDG target, and relatively few (28 out of 134) countries with electricity consumption above 350 kWh/capita do not meet the target.

The data presented here are indicative of the need for not only *access* to energy, but that there is apparently a *threshold* for developing human well-being and that the threshold goes beyond the definition for household "access" alone [17]. Although the identified thresholds corresponding to SDG3 vary somewhat between specific targets, the overall implication is that societal per capita average electricity consumption of less than a few hundred kWh has been historically inconsistent with being able to meet the SDG targets. Intra-societal distribution effects are also important, and thus disparities

within countries must also be quantified [11], as must other factors such as the institutional and policy structures in place within a country that can ensure effective implementation of electricity consumption in support of broader development goals. However, the dividing line shown here in terms of electricity consumption is remarkable even at this macro-level.

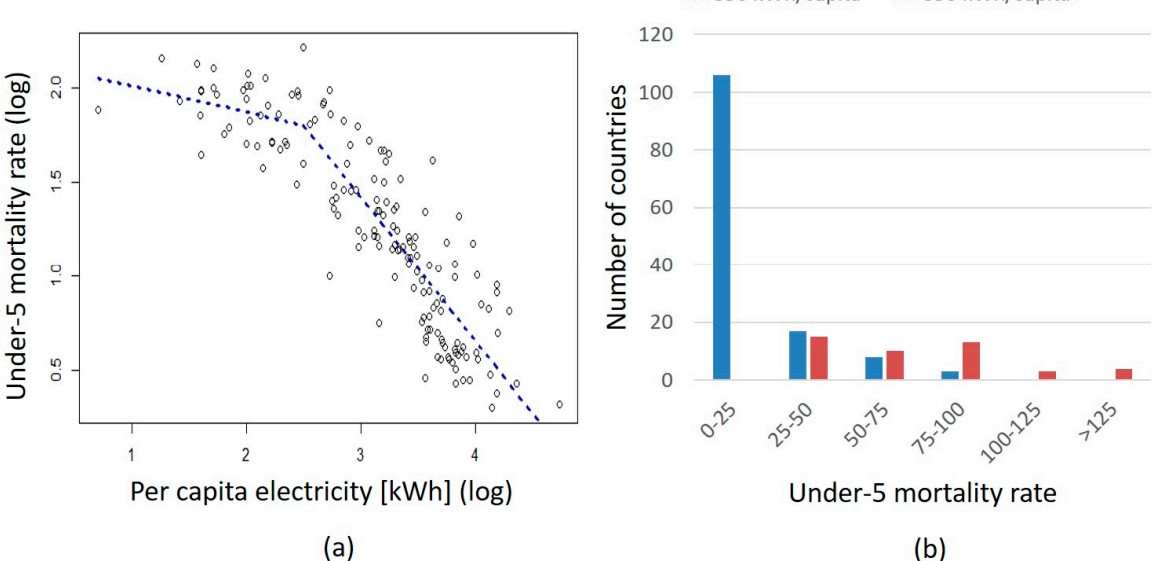

(a)                                                          (b)

**Figure 3.** (**a**) Threshold fitting of under-5 mortality ratio (log) vs. per capita national electricity consumption (log); (**b**) Histograms showing that for none of the countries with less than the indicative threshold electricity consumption of 350 kWh/capita, is the SDG3 target of reducing under-5 mortality rates to below 25 per 1000 live births met.

Although energy consumption may be key to achieving many SDG targets, simply making a supply of electricity available in a given country without any consideration of how access to modern infrastructure is translated into useful service outcomes is not a solution in and of itself. Further investigation is necessary to determine key energy inputs for improved health outcomes as one component of infrastructure needs in developing countries [27–30]. Energy access at clinics and hospitals, for example, is not the only necessary condition to be met, however. Connections between electricity consumption, food preservation, street lighting, water treatment and more are all part of contributing to health outcomes [16]. Here we have looked at health indicators because the SDG3 targets are well-quantified.

### 3.2. Projections of Future Electricity Needs for Meeting SDGs+

Countries with average per capita electricity consumption of less than 400 kWh per year have uniformly poor human development indicators, and do not currently meet SDG targets. It therefore appears that a reasonable quantitative target that would also correlate with enabling multiple sustainable development goals (where the emphasis here has been on specific health targets) would be to ensure consumption of at least 400 kWh per capita per year of electricity by 2030. (The threshold is indicative; it might be reasonable to choose 1 kWh per day or 365 kWh per year.) In fact, given the data presented above, an explanation would be required as to how any country would be able to meet the SDG targets with lower per capita consumption of electricity, since no country to date has been able to do so. As discussed in the introduction, simply consuming electricity is not a sufficient condition for development; the necessary accompanying condition would be that access and consumption of electricity be in productive sectors such as public facilities and infrastructure, in addition to individual households.

One of the regions most in danger of not meeting the SDGs is that of sub-Saharan Africa (SSA). By combining population changes, current electricity consumption, and the indicative target of 400 kWh per capita per year electricity consumption, it is possible to estimate total electricity needs in the SSA region by 2030, the target date for achieving the SDGs. The population in 2014 for those countries in SSA with less than 400 kWh per capita electricity consumption was 650 million. The UN projects in its "medium variant" scenario an increase in population to 1200 million in 2030 for these same countries [23]. Countries in SSA currently with less than 400 kWh per capita electricity consumption have an average of about 140 kWh per capita, and total electricity consumption for the 650 million people in these countries is therefore 91 TWh. Taking as a target that each of these countries should have an average per capita electricity consumption of 400 kWh by 2030 leads to an estimated total consumption of 480 TWh. Increasing regional total electricity consumption from the present level of 90 TWh to 480 TWh over a fifteen-year time period (where 2015 was used as the baseline year with available data) represents a compounded annual growth rate of 11%/year. A growth rate of 11%/year would on the one hand be unprecedented; over the past decade, India has been increasing electricity generation at over 6% per year, and China at over 8% per year [31]. On the other hand, renewable energy installations of wind power and solar photovoltaic capacity have grown at annual rates of 18%/year and 39%/year worldwide, respectively [31].

To put this growth rate in perspective, however, current world consumption of electricity is approximately 22,000 TWh [31], of which only 0.7% is consumed by the 13% of the world's population with consumption of less than the 400 kWh per capita under consideration here (and 40% of the world's population has consumption of less than 1000 kWh/capita, corresponding to a level at which almost all countries meet SDG health targets). In a Paris Agreement compatible climate change mitigation scenario, total electricity consumption is expected to increase to about 35,000 TWh by 2030, depending on the model [32]. In the SDG and energy-consumption-driven formulation presented here, of the additional 13,000 TWh projected for worldwide consumption compared to today, having 3% of that *increase* go to 13% of the world's population might enable the success of several SDGs simultaneously.

Recent reports show that power generation in all of Africa is expected to double [32] or even triple [33] between 2010 and 2030, to about 2000 TWh in the latter case, from 650 TWh in 2010. If these increases in electrification, are targeted toward the poorest countries such as to lift per capita consumption above the 400 kWh level by 2030, this represents about one-third of the continent's total increase in electricity generation. The estimated investment necessary to achieve this partial step is between $150–250 billion over 15 years [5,33].

Another recent report examines the decreasing costs of mini-grids [34]; the additional electrification postulated above for electricity-poor countries might be achieved for roughly $200 billion over the same period, which may be a preferred solution given the preponderance of lack of electricity access in rural areas where mini-grids may be most suitable. The investment would be for infrastructure that has a lifetime well beyond the fifteen years under consideration for the SDGs, and given synergies between energy consumption and other activities that further development indicators, would serve as leverage for increasing well-being.

## 4. Discussion

Enabling sustainable human development for all will require access to modern energy technologies. Whereas past human development was enabled by fossil fuels, mitigating the most harmful manifestations of climate change and their disproportionate negative impacts on developing countries will require new sources of near-zero-carbon technologies to be introduced over the next few decades, both in industrialized countries and especially at a much increased rate in those countries still on the path of development.

The technologies to enable sustainable development through renewable energy are either currently available or advancing rapidly. The cost for installation of solar photovoltaic systems has decreased by more than 80% in the past decade and the cost of wind power has also decreased significantly [35–37].

Ten years ago, or even five years ago, legitimate questions were asked as to how those in wealthy countries could expect developing countries to invest in expensive renewables when they themselves were not doing so. Since then, investment trends in wealthier countries have been leaning toward renewables and away from fossil-fuel energy [38]. Increasingly, an economically consistent case can be made that enabling energy access for developing countries means ensuring access to sustainable, renewable energy, as expressed in SDG7, Target 2.

At the same time, there is a need to reconcile other SDG targets with the single indicator of energy consumption. If the SDGs are to be met by 2030, and if more than a minimal commitment to reducing poverty and increasing access to health care and to education is the real aim, then access to modern energy systems appears to be a pre-requisite. The amount of energy needed will vary by world region, but there is a threshold, perhaps as low as 400 kWh per person per year for the whole society. In effect, the view taken here is that energy consumption be used as a quantifiable leverage point for meeting SDGs, and that at the same time, and assuming that much of that additional electricity will be from renewable sources, the Paris Agreement and SDG 13 can also be supported. In fact, there is at least some preliminary investigation into a negative feedback loop scenario in which meeting the SDGs will in itself lead to a decrease in population growth, thus further reducing pressures on both earth and human systems [39].

Not all SDG indicators show clear threshold behavior, but they can often be tested in a similar way to determine if targets are consistent with electricity consumption levels. Examples are shown in the Supplementary Information with respect to SDG6 is to "ensure access to water and sanitation for all," [7] and SDG2, "End hunger, achieve food security and improved nutrition and promote sustainable agriculture."

Furthermore, one possible objection to the analysis presented here is that we have used electricity consumption as the independent variable, but other variables could have been chosen just as easily. In other words, energy consumption could be simply a proxy for "wealth" or GDP/capita, for example. Although the two are closely related, as shown in the Supplementary Information, the threshold behavior described above in terms of electricity consumption is not evident as a function of GDP.

The Intergovernmental Panel on Climate Change (IPCC) Fifth Assessment Report (AR5) [40] and the Special Report on Global Warming of 1.5 °C [41] conclude that impacts of climate change and vulnerability to those impacts will be felt most strongly by the poorer nations of the world. This conclusion is not surprising, since developing countries, almost by definition, do not have the robust infrastructure necessary to withstand additional stressors. The direct and often disproportionate impact of climate change on developing countries is recognized by SDG13.

Several of the SDGs can be conceived of as co-requisite, or at the very least, mutually-enabling of one another [18,19,21,42]. These conditions are not automatically satisfied, however, and one area for potential conflict between goals might be precisely between SDG13 and others. For example, an argument is sometimes made that fossil-fuel energy is necessary for poorer countries; building up such infrastructure, however, would be inimical to efforts to mitigate climate change impacts [43]. Attention to possible conflicts between SDGs is needed, if for no other reason than to find alternatives that avoid dead-ends in development or climate policies.

We emphasize that studies at higher spatial, energy system and socioeconomic resolution are needed to determine the most effective means of leveraging energy access at different levels to achieve SDG targets. The analysis presented here is not intended to imply that a certain amount of electricity consumption will guarantee SDG targets, or that all of that electricity consumption will be in, for example, the health sector. Rather, energy (and specifically, sustainably generated electricity) infrastructure enables a complex chain of feedback and positive influences that on a societal level make meeting SDG targets significantly more likely.

The results presented here are meant as a starting point for quantifying SDG energy targets by referencing other aspects of sustainable development. Social, cultural, political and economic factors will play a critical role in determining if a successful transition to a sustainable energy system can be

made, while simultaneously reducing impacts on the climate system, and fulfilling the Sustainable Development Goals. Here we have shown a strong indication that only with significant levels of electricity consumption will it be possible to meet SDGs, or at least that if there is to be development without consumption of electricity at significant levels, a dramatic break with past history, it will be necessary to understand how this might be brought about.

**Supplementary Materials:** The following are available online at http://www.mdpi.com/2071-1050/11/18/5047/s1. 1. Data sources; 2. Details on Threshold fits and statistics for SDG3-related targets; 3. Energy access and human development; 4. HDI and electricity consumption; 5. Electricity consumption thresholds for further SDG targets 6. Threshold analysis dependent on chosen independent variable?

**Funding:** The authors acknowledges and appreciates funding by the German Federal Ministry for the Environment, Nature Conservation and Nuclear Safety (16_II_148_Global_A_Impact) and from the University of Dayton for sabbatical leave support.

**Acknowledgments:** R.J.B. wishes to thank colleagues and students at the University of Dayton in the Hanley Sustainability Institute and the Human Rights Center for encouragement and feedback. R.J.B. also wishes to thank Climate Analytics for hosting him during a sabbatical stay and colleagues there for critiques of this work.

**Conflicts of Interest:** The author declares no conflict of interest.

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
