# Peer review of "Threshold Electricity Consumption Enables Multiple Sustainable Development Goals"

_sustainability, doi:10.3390/su11185047_

Round 1
Reviewer 1 Report
Comments are in the attached document.

Author Response
From the pdf document I have extracted the Reviewer's comments and answer these here:
(abstract) Mention the methodology used This has now been done (briefly) in the abstract (Introduction) Source missing Added a general reference (Introduction) What is the source of this particular argument? This part of the introduction has been rewritten to enhance clarity, also with respect to comments by another ReviewerReviewer 2 Report
Thank you for the opportunity to review this paper. The content is of high importance and interest to the readership. However, the arguments developed are too simplistic and several flaws prevent this paper from being published in its current state.
The reasons why I recommend that this paper be published after major revisions are as follows:
· Fundamentally, the paper correlates per-capita electricity consumption with SDG3 indicators, and uses this correlation to come to the conclusion that a certain minimum level of per-capita electricity consumption should be set as a quantifiable target for SDG7.1. While there may be merit in revisiting the existing definition of electricity access for SDG7.1 (a single electricity connection), there is a rich literature and background to this target, which the author does not engage with and in fact mis-represents on several occasions (more below). The justification the author uses to circumvent “correlation is not causation” on line 58 is not clear. What the author is trying to conclude, my interpretation is, is that the current definition of electricity access is not sufficient to make meaningful contributions to development indicators. However, their suggestion to redefine SDG7.1.1 to electricity consumption is not justified by the analysis.
· “Access to” 400kWh of electricity per year, neglects how this electricity is used (what sectors and for what purposes), the distribution of consumption across the population, what technologies provide that electricity generation (efficiency, air pollution, reliability, affordability).
· A discussion of the interlinkages between energy access and health is missing.
· Only electricity is considered, rather than energy as a whole, including all modern fuels. Why is this decision taken?
· The SDG7 target refers to universal access to modern, clean, affordable and reliable energy for all. Electricity access is just one indicator (SDG7.1.1), but there are three other indicators, including access to clean cooking fuels, renewable energy and energy efficiency. All indicators, other than 7.2.1 (renewable energy) are quantifiable, in contradiction to the author’s assertion (line 100, line 251). This is a very basic error to make.
· Throughout the paper, the author refers to “population with access to electricity” to mean “population living in a country with per-capita electricity consumption less than x kWh”. This usage is very non-standard and should be revised.
· The conclusions reached from Section 3.2 are not elaborated on in sufficient detail. What is the purpose of quantifying additional electricity demand? The sentence from line 204 is not understandable. The comparison with the IEA Energy for All case is not elaborated on – this analysis should be either fully fleshed out (not in SI) or removed. It appears that the author is comparing achieving universal energy access using this new threshold method with the traditional “connections” analysis.
· There is a misunderstanding of the IEA definition of energy access. Please see the following: https://www.iea.org/media/publications/weo/EnergyAccessOutlook2017Definingandmodellingenergyaccess.pdf “The IEA defines energy access as "a household having reliable and affordable access to both clean cooking facilities and to electricity, which is enough to supply a basic bundle of energy services initially, and then an increasing level of electricity over time to reach the regional average". This energy access definition serves as a benchmark to measure progress towards goal SDG 7.1 and as a metric for our forward-looking analysis.”
· In the abstract (from line 21) and elsewhere, this threshold is confused with IEA definitions. The 250/500kWh per household per year is an initial level of household electricity consumption in the first year after gaining access – the consumption grows over time to meet the regional average; and this is not a definition of access but used for modelling purposes. This is also just household electricity consumption for those gaining access. The author’s 400kWh per capita per year is country-average for all people and sectors. Therefore the comparison is not valid.
· The term “breakpoint” is not great – how about “threshold”?
· No treatment of the very poor quality of energy statistics in developing countries, especially Africa.
· The figures 1-3 – why log scale? The Figures without log scale from SI with SDG3 targets would be interesting.
· Please define the SDG7 and 3 goals, targets and indicators.
I find the analysis in parts very interesting, and has the potential to add to the literature around defining energy access and interlinkages with other SDGs. The paper is also well written. I find, however, that the execution and conclusions are not adequate in their current state and therefore recommend that paper be revised and resubmitted under those conditions.
Round 2
Reviewer 2 Report
I thank the author for the extensive revisions in response to my review. Considering these revisions, particularly the focus on electricity consumption rather than electricity access greatly enhances the paper and therefore I would be happy to recommend that it be published in its current form.